# Validation and Normative Data on the Verbal Fluency Test in a Peruvian Population Ranging from Pediatric to Elderly Individuals

**DOI:** 10.3390/brainsci12121613

**Published:** 2022-11-24

**Authors:** Jonathan Adrián Zegarra-Valdivia, Brenda Chino, Carmen Noelia Paredes Manrique

**Affiliations:** 1Faculty of Health Sciences, Universidad Señor de Sipán, Chiclayo 14001, Peru; 2Medicine Faculty, Institute of Neuroscience INc, Universidad Autónoma de Barcelona, 08193 Barcelona, Spain; 3Psychology Department, Universidad Nacional de San Agustín de Arequipa, Arequipa 04001, Peru

**Keywords:** verbal fluency, normative, neurocognition, executive function, pediatric, elderly

## Abstract

In neuropsychological evaluation, verbal fluency is a crucial measure of cognitive function, but this measure requires standardized and normative data for use. The present study aimed to obtain validation and normative data for the verbal fluency task in the Peruvian population, with participants ranging from 6 to 94 years and varying in age, educational level, and sex. We recruited 2602 healthy individuals and used linear regression analysis to determine the effect of age, sex, and educational level. We also evaluated internal consistency between categories and phonological tasks with Cronbach’s alpha and Pearson’s correlation analysis and calculated test-retest reliability after three months. We found significant effects of age, educational level, and sex on phonological and semantic fluency. Participants with more than 12 years of education had the highest scores overall. Regarding age, middle-aged participants (between 31 and 40 years old) had the highest scores; scores gradually decreased outside of this age range. Regarding sex, men performed better than women. These results will increase the ability of clinicians to precisely determine the degree to which verbal fluency is affected in patients of different ages and educational levels.

## 1. Introduction

Verbal fluency (VF) tasks are a group of neuropsychological assessments widely used in clinical practice and research. These tasks consist of naming as many words that follow a series of orthographic or semantic rules as possible in a given period (usually 60 s) [1]. Each list of words follows a specific criterion, such as starting with a particular letter (phonological VF) or mentioning words in a category (semantic VF) [2,3].

Despite the apparent similarity between these two types of tasks due to the use of language as a critical component [4] and search strategies through memory [5,6], neuroimaging studies have indicated that they utilize different underlying brain circuits. Phonological fluency is related to greater activation of the frontal lobe and executive function [4], while semantic fluency is an alternative to the lexicon and lexical access and requires activation of the temporal lobe [7,8,9].

In addition, the VF tests are quick and easy to administer and sensitive to cognitive impairment in a variety of disorders [1], facilitating the detection of early stages of neurodegenerative diseases such as mild cognitive impairment, Alzheimer’s disease [10,11], Huntington’s disease [12], attention-deficit/hyperactive disorders [13], traumatic brain injury [14] and aphasia [15].

VF performance is evaluated by recording the total number of words produced during the task and the number of words retrieved during individual portions. During the first 30 s, healthy subjects provide approximately two-thirds of their total words, followed by a drastic increase in the retrieval period [14,16] because the effort required to produce words increases with time, necessitating progressive increases in attention and executive control as the task continues.

The interpretation of VF results may differ from that of the normative data used for comparison [16]. However, the cognitive processes involved in the evaluation are similar. Phonological fluency is widely interpreted as a strategic search-and-retrieval measure within orthographic or phonological networks that involves a series of higher-order functions (working memory, inhibition, and alternation). Other research has highlighted substantial contributions of verbal intelligence and information processing speed to phonological fluency in healthy populations [1]. For semantic fluency, subjects must generate and follow a strategy to efficiently explore the semantic network. In general, healthy subjects exploit the internal organization of this network to explore a semantic category (for example, fruits). Then, they flexibly move between different subcategories or elements to select one of the available options. Afterward, the subjects must extract entries from semantic memory and monitor and verify the output to avoid repetitions or out-of-category responses. Finally, subjects must maintain an “active” state during task execution to address the limited time available for production [11].

Other features related to VF performance include vocabulary size, lexical access speed, updating, and inhibition, which are mainly associated with the speed of the first responses [17]. Semantic VF has been assessed in more than fifteen languages, including Indo-European, Semitic, Sino-Tibetan, Austroasiatic, Dravidian, and even Amerindian languages [18]. Similarly, phonological VF has been assessed in different languages, including Spanish, regardless of participants’ first language and ethnic background, and is helpful for diagnostic purposes [19].

Although cognitive evaluation is a crucial part of the clinical approach in neuropsychology or clinical psychology, in the Peruvian context, many important instruments, such as the VF test, are not standardized or lack normative data. Furthermore, test performance is usually influenced by sociodemographic variables such as age, educational level, and sex [3,16,20].

Several studies have documented the clear need to obtain normative data from a country to interpret the results of neuropsychological tests [21,22]. According to Hazin et al. [23], significant differences in children’s performance on formal academic performance tests among regions are observed only in developing countries [22]. Additionally, a review by Ramírez et al. [24] suggested possible cultural effects, in addition to variables such as age and schooling, on the VF results obtained in Hispanic samples; however, their results were contradictory.

Despite sharing the same language, Hispanic countries may differ in the quality of education, which could generate masking patterns unique to a particular ethnic group [25]. Professionals should be cautious when applying qualification standards because of the variability in demographic factors among geographic regions, which can interfere with performance [26]. In Peru, a large and regionally diverse country, normative data are needed considering the variation in a number of factors, including languages spoken (monolingual or bilingual), residential area (urban or rural), and educational level (including illiteracy). A recent contribution [8] obtained normative data for the VF test from eleven countries in Latin America, including Peru. However, the age range spanned only 6 to 17 years; thus, these data remain insufficient.

Given the above factors, it is necessary to determine specific parameters of the Peruvian population to determine the influence of different demographic variables on VF performance. This study aimed to obtain validation and normative data on the VF task in the Peruvian population ranging from 6 to 94 years of age, accounting for age, educational level, and sex.

## 2. Materials and Methods

### 2.1. Participants

An initial sample of 3524 individuals was recruited from Arequipa, Lima, and Chiclayo, Peru. After applying the inclusion and exclusion criteria, the final selection consisted of 2602 healthy participants. The mean ages in this study ranged from 6 to 94 years, and 55.3% of participants were female. Participants were selected according to the following criteria (which vary according to age group): (1) verbal or written consent to participate provided by the subject or caregiver, legal guardian, or another proxy; (2) IQ > 85 as evaluated with the computerized version of the Raven progressive matrix test or the test of nonverbal intelligence (TONI version II); (3) absence of cognitive impairment (in older people) as indicated by a Mini-Mental State Examination (MMSE) score ≥ 24; (4) without depression, as determined by scores on the Children’s Depression Inventory (CDI; children), the Hamilton Depression Scale (young and middle-aged individuals), or the Geriatric Depression Scale (GDS; elderly individuals); (5) no history of neurological or psychiatric disease according to clinical history or psychological assessment; (6) no sensorimotor or language impairment; and, (7) use of Spanish as their primary language or an extensive history of speaking Spanish (more than 20 years). Further details are provided in Figure 1.

Participants were recruited from public and private schools, a technological institute, and senior centers from Arequipa, Lima, and Chiclayo. After obtaining approval from the specific institution, subjects were informed about the study’s purposes and provided verbal or written consent. The present study was an instrumental study [27].

### 2.2. Testing Procedure

Subjects were tested individually in a quiet room in their specific location (school, institute, or clinic). The sequence in which tests were administered was identical for all subjects. The procedure included two to three sessions (almost two hours). Participants were tested at 10 am and provided 15 min to relax between sessions. We used letters (phonological fluency: F-A-S-M-R-P) and categories (semantic fluency: animals and fruits) in the VF test because these rules are the most studied in the literature [3,4]. All participants were native Spanish speakers. Non-native Spanish speakers were not included in this study.

Participants were given the following instructions to assess phonological fluency (representative examples provided): “I am going to say a letter of the alphabet, and I would like you to say as many words as you can think of that start with that letter, excluding proper nouns (i.e., names of people or places). Are you ready? You have one minute, and the letter is P.” “Now we will try a different letter. Similar to the previous task, please say as many words as you can think of that start with the new letter, avoiding proper nouns (i.e., names of people or places). The new letter is F.”

To assess semantic fluency, participants were provided with the following instructions (representative example provided): “Now, please name as many animals as you can that start with any letter. Again, you have one minute. Start now”.

### 2.3. Verbal Fluency Scoring

We recorded the total number of responses on the phonological or semantic fluency tasks. To calculate the raw score of a participant, we awarded one point to each correct answer and excluded any repetitions or derivative responses (diminutive or augmentative responses). Errors were classified as perseverations or intrusions.

### 2.4. Group Stratification

We stratified our groups by age, sex, and educational level (shown in Appendix A), considering the Peruvian educational system. The first group consisted of children between 6 and 8 years old, as Peruvian children acquire and consolidate reading and writing skills in primary school (first and second grade). The second group consisted of children from 9 to 11 years old (from third to sixth grade), at which point children are typically fully literate. The third group consisted of children from 12 to 14 years old (in secondary school, from first to third grade). The fourth group consisted of adolescents 15 to 17 years old (finishing secondary education, from fourth to fifth grade). The fifth group consisted of individuals 18 to 20 years old; at this age, young Peruvians usually attend university or pursue technical education. After this first division, we subsequently grouped individuals according to age from 21 to 90 years old, as we observe no differences between the percentiles of the groups. These age groups allowed larger groups at specific ages. We also present descriptive statistics and percentiles for males and females to assess sex differences. Educational level was divided into the following three categories: between 1 and 6 years of education (primary school), between 7 and 11 years (secondary school), and more than 12 years (technical school or university). We believe that our stratification system allows more realistic and ecological assessments of Peruvian VF performance.

### 2.5. Ethical Statement

The study complied with the ethical considerations related to clinical trials, and all methods were performed according to the relevant guidelines and regulations of the Declaration of Helsinki. All participants were informed about the aims and risks of this study and provided written or verbal informed consent. For minors, parents provided informed consent. Institutional approval was obtained from each institution (public and private schools, regular primary education: IE Florentino Portugal, IE San Pablo, IE Miguel Grau; secondary education: IES San Jose; and health centers–Peru Ministry of Health [MINSA]). In addition, a Local Research Ethics Committee (Neuroscience Group Ethics Committee; CEI number 001-2020) approved the study. All data were collected in an anonymous database.

### 2.6. Data Analysis

The sociodemographic characteristics of the participants included in the study were compared with *t* tests and chi-square tests. A linear regression analysis assessed the effect of age, sex, and educational level. Performance significantly differed according to educational level and age. No effect of sex was found after adjusting for age and educational level. We investigated different age groups, ranging from 6 to 94 years old (every five years).

Nine educational level groups were considered. Therefore, the sample was stratified according to age, educational level, and obtained percentile (Table 1). These effects were assessed by multivariate analysis of variance (MANOVA). Additionally, Cronbach’s alpha was calculated to evaluate internal consistency and intraclass correlation coefficients (ICCs) were calculated between categories and phonological tasks to assess the reliability of each measure. We also performed a Spearman correlation analysis with a subsample (*n* = 179) who underwent the same protocol after three months (post-test; semantic category: animals, phonological letter: P) for test-retest reliability. Statistical analysis was performed with SPSS version 24 (SPSS, Chicago, IL, USA). Significant results are indicated with * *p* < 0.05 and ** *p* < 0.01.

## 3. Results

We present the results of 2602 healthy participants. The total correct phonological and semantic VF responses were first calculated for statistical analysis. Table 1 shows the mean, standard error of the mean (SEM), standard deviations (SDs), perseveration errors and perseveration rates for each letter and semantic category. Correct answers for each letter according to educational level (three categories) are shown in Table 2.

We found significant effects of age, educational level, and sex (Table 3) on phonological and semantic fluency, as revealed by MANOVA. The highest scores in phonologic fluency were from participants with more than 12 years of education. In contrast, middle-aged participants (between 31 and 40 years old) had the highest scores on semantic fluency; scores gradually decreased outside of this age range. Males exhibited better performance than females.

Pearson correlation analysis was performed and ICCs were calculated (Table 4) for each letter category in phonologic fluency and both semantic categories (animals and fruits) for semantic fluency. As shown in Table 4, the correlations between letter performance (F–A–S–M–R–P) ranged from 0.693 to 0.863. The correlation of semantic fluency (between the two semantic categories) was 0.690. The ICCs for phonologic and semantic fluency were 0.954 and 0.811, respectively. Moreover, test-retest reliability was evaluated with Spearman correlation analysis (Table 5) in a subsample of participants (*n* = 179). There was a significant correlation between pre- and post-test performance (rho = 0.3.666, *p* = < 0.001 **). Normative data are presented in Appendix A (descriptive statistics and percentile tables). We considered different percentiles regarding educational level, which had the most impact on VF performance.

Finally, a linear regression was performed to verify the relationship of phonologic (R^2^ = 0.187) and semantic fluency (R^2^ = 0.076) with age, sex, and educational level (see Table 6 and Table 7). The variables selected explained almost 18% of the variance in phonologic fluency and 7% of the variance in semantic fluency. We believe that our sample is representative and unlikely to be conditioned or influenced by the sociodemographic variables studied. Thus, 82% of the variability in phonological fluency and 93% of the variability in semantic fluency may be explained by other cognitive variables, such as processing speed, working memory, and executive function.

## 4. Discussion

This study specifically attempted to obtain normative data on the neuropsychological VF test in the Peruvian population. We recruited more than 2600 healthy native Spanish speakers. These individuals varied in educational level and age, ranging from 6 years to 94 years (seventeen age groups). Percentiles were obtained for age, educational level, and sex (see Appendix A).

Education directly influences performance on several neuropsychological tests and modifies the brain’s functional organization after exposure to reading and writing [3]. We found that participants with more than twelve years of education had better scores on each letter category (phonological fluency) and semantic category (semantic fluency). Thus, the years of education are highly correlated with performance on this test [3,25]. Ratcliff et al. [28] reported that educational level influenced phonological fluency more than semantic fluency and that participants with fewer years of education generated fewer words. However, according to Ostrosky-Solis et al. [3], age is the most robust predictor of verbal fluency in highly educated people (with >10 years of education). Nonetheless, the total semantic fluency score of individuals with 0 to 4 or 5 to 9 years of education is most strongly influenced by educational level without a significant contribution of age. This effect may be due to the educational ranges included in most studies, which range from participants with little or no education to those with up to 8 years of formal education [29].

Our results regarding the influence of age on performance are similar to those of other studies. Previous studies have shown that increased age is associated with significant decreases in incorrect words and increases in repeated words [6,14]. Phonological fluency usually exhibits a curvilinear relationship with age, with an increase in fluency between the third and fourth decade followed by a gradual decrease. In contrast, semantic fluency shows a linear decline with age [4]. The semantic advantage persists until the eighth decade of life [6]. Generally, phonological VF requires more elaborate organization and retrieval strategies than semantic VF [30]; thus, these differences in difficulty persist throughout life [2].

In addition, previous studies have reported inconsistent effects of sex on VF performance. Many studies have not detected significant sex differences, while others have found that women exhibited superior performance [4]. Our data indicate a male advantage, in contrast to the results obtained by Mitrushina et al. [31] or Vaughan et al. [6], which shows a significant effect of sex on F-A-S performance, with better performance exhibited by women. In our study, men performed slightly worse on the phonological test (using the letter “F”) but produced, on average, 0.5 more examples than women on the semantic fluency test (animal category). These sex differences are further complicated by sex differences in familiarity with specific semantic categories.

The present study provided validation and normative data on the performance of people between 6 and 94 years of age on phonological and semantic fluency tasks; such data from Peruvians were previously lacking. Our results will increase the ability of clinicians to precisely determine the severity of VF impairment in patients with different ages and educational levels to make differential diagnoses of other disorders.

Nonetheless, this study have some limitations such as non-representative population of rural or other languages from Peru, or relatively low explained variance in the regression analyses; the fact that the results could not be generalized to individuals outside Peru and some groups were reduced.

## 5. Conclusions

Educational level directly influenced VF during schooling and was highly correlated with VF performance. In our study, participants with more than twelve years of education performed better on each letter category (phonological fluency) and semantic category (semantic fluency).

Age-based changes in phonological fluency typically assume a curvilinear pattern, with an increase in phonological fluency between the third and fourth decades, followed by a gradual decline. In contrast, semantic fluency exhibited a linear decrease with age. The semantic advantage persists into the eighth decade of life. In addition, previous studies have reported inconsistent effects of sex on VF performance. We found that men exhibited better performance. Although male performance was slightly worse on the phonological task (using the letter “F”), male participants produced, on average, 0.5 more examples than women on the semantic fluency task (animal category). These sex differences are further complicated by sex differences in familiarity with specific semantic categories.

## Figures and Tables

**Figure 1 brainsci-12-01613-f001:**
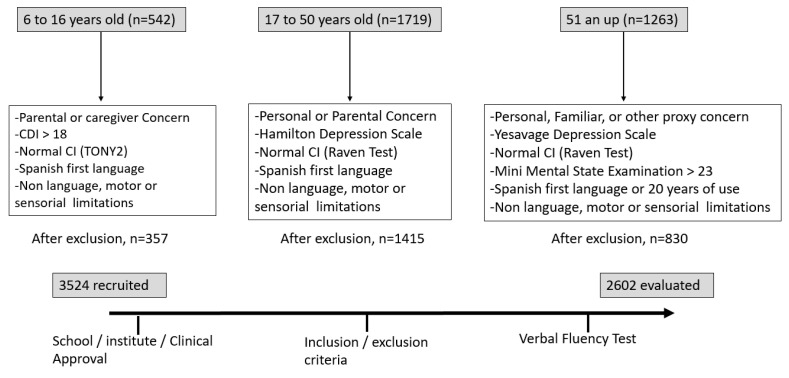
Flow chart of participant recruitment and the final study sample obtained after applying the inclusion/exclusion criteria that involved different tests and psychological evaluation.

**Table 1 brainsci-12-01613-t001:** Phonologic and semantic verbal fluency of all healthy participants (*n* = 2602).

Verbal Fluency	Perseveration Error	Perseveration Rate	Mean	SEM	SD
F	-	-	8.97	0.094	4.773
A	-	-	9.87	0.08	4.071
S	-	-	9.13	0.083	4.217
M	-	-	9.71	0.086	4.393
R	-	-	9.43	0.085	4.334
P	-	-	11.1	0.087	4.447
Animals	-	-	15.54	0.1	5.101
Fruits	-	-	12.96	0.086	4.386

**Table 2 brainsci-12-01613-t002:** Phonologic and semantic verbal fluency of all healthy participants according to educational level (years).

Educational Level (Years)	1–6			7–11			>12		
	Mean	SEM	SD	Mean	SEM	SD	Mean	SEM	SD
F	5.8	0.172	3.724	9.61	0.108	4.693	10.46	0.301	4.529
A	7.51	0.178	3.839	10.18	0.087	3.772	12.44	0.303	4.558
S	6.01	0.17	3.678	9.79	0.09	3.9	10.46	0.307	4.63
M	6.21	0.171	3.706	10.52	0.093	4.064	10.55	0.304	4.585
R	6.04	0.171	3.705	10.17	0.092	4.019	10.53	0.3	4.522
P	7.87	0.174	3.765	11.78	0.096	4.192	12.46	0.307	4.625
Animals	12.81	0.243	5.25	16.26	0.108	4.701	15.57	0.388	5.85
Fruits	11.38	0.237	5.116	13.05	0.086	3.745	15.66	0.393	5.926

**Table 3 brainsci-12-01613-t003:** Total phonologic and semantic verbal fluency scores according to educational level (years), age range (years), and sex. Multivariate analysis of variance (MANOVA): *p* < 0.05 *, *p* < 0.01 **.

		Phonologic Fluency		Semantic Fluency	
Category	N	M	SD	M	SD
Educational level, years					
1–6	482	6.5753	3.57927	12.0964	4.94653
7–11	1893	10.3414	3.52154	14.6574	3.7629
>12	227	11.149	4.54684	15.6145	5.86331
F		159.441		60.968	
*p* value		<0.001 **		<0.001 **	
Age range, years					
6–8	68	5.2181	1.9084	10.3015	2.69485
9–11	77	7.4069	2.45299	12.961	3.08942
12–14	98	10.716	3.16908	15.6684	3.08132
15–17	369	11.2168	2.87716	15.1843	3.18303
18–20	626	10.4846	3.06305	14.6973	3.17059
21–25	378	11.082	3.27878	14.9299	3.3389
26–30	63	11.0556	3.26969	15.1746	3.00558
31–40	29	11.7989	2.52895	14.3966	2.9197
41–50	64	9.3229	5.56918	14.4297	6.40428
51–55	168	9.8423	4.66468	14.3304	5.58276
56–60	187	8.6934	4.53526	14.361	5.53842
61–65	143	8.0921	4.53985	13.507	5.83034
66–70	145	7.2736	4.24985	12.8655	5.92261
71–75	66	7.4899	3.91347	12.0758	5.40814
76–80	54	6.642	3.94171	12.6296	5.7897
81–85	36	6.588	3.39923	11.5139	5.4635
>86	16	5.7564	3.49317	10.96795	3.965395
F		32.854		11.48	
*p* value		<0.001 *		<0.001 *	
Sex					
Male	1163	9.8719	3.96596	14.459	4.31937
Female	1439	9.5609	3.87762	14.0578	4.37206
T		3.278		4.341	
*p* value		0.070 *		0.037 *	

**Table 4 brainsci-12-01613-t004:** Pearson correlation coefficients and interclass correlation coefficients between phonologic and semantic fluency. *p* < 0.01 **.

	F	A	S	M	R	P	Animals	Fruits
F	1	0.718 **	0.710 **	0.729 **	0.720 **	0.693 **	0.445 **	0.384 **
A	0.718 **	1	0.789 **	0.787 **	0.778 **	0.769 **	0.486 **	0.473 **
S	0.710 **	0.789 **	1	0.833 **	0.828 **	0.795 **	0.513 **	0.424 **
M	0.729 **	0.787 **	0.833 **	1	0.863 **	0.831 **	0.542 **	0.417 **
R	0.720 **	0.778 **	0.828 **	0.863 **	1	0.828 **	0.507 **	0.411 **
P	0.693 **	0.769 **	0.795 **	0.831 **	0.828 **	1	0.513 **	0.416 **
Animals	0.445 **	0.486 **	0.513 **	0.542 **	0.507 **	0.513 **	1	0.690 **
Fruits	0.384 **	0.473 **	0.424 **	0.417 **	0.411 **	0.416 **	0.690 **	1
ICC Cronbach’s alpha	0.954						0.811	

**Table 5 brainsci-12-01613-t005:** Spearman correlation coefficients for phonologic and semantic fluency between pre- and post-test performance. *p* < 0.05 *, *p* < 0.01 **.

			Post-Test							
			Animals	Fruits	F	A	S	M	R	P
Pre-test	Animals	Coefficient	0.366 **	0.259 **	0.287 **	0.283 **	0.170 *	0.322 **	0.224 **	0.287 **
		*p* value	<0.001	<0.001	<0.001	<0.001	0.023	<0.001	0.003	<0.001
	Fruits	Coefficient	0.288 **	0.571 **	0.290 **	0.362 **	0.122	0.259 **	0.331 **	0.263 **
		*p* value	<0.001	<0.001	<0.001	<0.001	0.105	<0.001	<0.001	<0.001
	F	Coefficient	0.307 **	0.247 **	0.515 **	0.541 **	0.436 **	0.547 **	0.500 **	0.512 **
		*p* value	<0.001	0.001	<0.001	<0.001	<0.001	<0.001	<0.001	<0.001
	A	Coefficient	0.333 **	0.262 **	0.477 **	0.523 **	0.444 **	0.574 **	0.481 **	0.560 **
		*p* value	<0.001	<0.001	<0.001	<0.001	<0.001	<0.001	<0.001	<0.001
	S	Coefficient	0.308 **	0.160 *	0.435 **	0.459 **	0.449 **	0.518 **	0.398 **	0.484 **
		*p* value	<0.001	0.032	<0.001	<0.001	<0.001	<0.001	<0.001	<0.001
	M	Coefficient	0.301 **	0.280 **	0.523 **	0.530 **	0.474 **	0.643 **	0.520 **	0.616 **
		*p* value	<0.001	<0.001	<0.001	<0.001	<0.001	<0.001	<0.001	<0.001
	R	Coefficient	0.236 **	0.14	0.521 **	0.576 **	0.459 **	0.555 **	0.512 **	0.530 **
		*p* value	0.001	0.061	<0.001	<0.001	<0.001	<0.001	<0.001	<0.001
	P	Coefficient	0.334 **	0.243 **	0.559 **	0.617 **	0.464 **	0.643 **	0.478 **	0.721 **
		*p* value	<0.001	0.001	<0.001	<0.001	<0.001	<0.001	<0.001	<0.001

**Table 6 brainsci-12-01613-t006:** Results of simple linear regression analysis to verify the model. *p* < 0.01 **.

Verbal Fluency	R^2^	S.D.E. (Residual)	*p* Value
Phonological	0.187	3.559	<0.001 **
Semantic	0.076	4.196	<0.001 **

**Table 7 brainsci-12-01613-t007:** Results of simple linear regression analysis to verify associations of verbal fluency with age range, educational level, and sex. *p* < 0.05 *, *p* < 0.01 **.

Phonological			Semantic	
Variable	β	*p* value	β	*p* value
Age range	−0.234	<0.001 **	−0.120	<0.001 **
Educational level	0.349	<0.001 **	0.238	<0.001 **
Sex	−0.053	0.003 *	−0.050	0.008 **

## Data Availability

The data used and analyzed during the current study are available from the corresponding author upon reasonable request.

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
