# Peer review of "Validation and Normative Data on the Verbal Fluency Test in a Peruvian Population Ranging from Pediatric to Elderly Individuals"

_brainsci, 2022, doi:10.3390/brainsci12121613_

Round 1

Reviewer 1 Report

The objective of the present study is relevant for both research and applied fields of neuropsychology. In this regard the present work has the potential to improve psychological and neuropsychological assessment providing a new and representative set of norms of verbal fluency tasks for the Peruvian population. Moreover, the authors present novel data about internal consistency and test-retest reliability of the test relevant for many researchers in the field. Whereas the rationale and general justification of the study is clear, there are some confusion about the data, the analyses being performed, and the justification of those analyses that need to be clarified before the manuscript could be considered for publication.

Major points

I was unable to find percentile tables as reported in p. 4 lines 137-138 and in p.5 line 172. The authors state “Nine educational level groups were considered. Therefore, the sample was stratified according to age and education and obtained percentiles (Table 1).” However, the contents of Table 1 (p.4) doesn’t match this information. Table 1 is also mentioned within text in line 150 (apparently correct) referring descriptive statistics. Again in p.5 line 172, there is a reference to “supplementary Table 1” with percentiles that however are not reported in the supplementary Table 1 provided by the authors (with the title “Supplementary Table 1. Total scores for phonologic fluency, distributed by age and educational years, until 41 years old”). Please, clarify all this information and provide the editor percentile tables before the contents of the manuscript could be properly reviewed.

In any case, and once this confusion became clarified reconsider whether “supplementary materials” is the appropriate section to report the main result of the present investigation, that is, the norms.

I have serious doubts about the necessity, utility, and representativity of all the groups being stratified to generate the present norms. Even with such a big sample, the stratification of groups should be grounded on an appropriate justification. Moreover, I’m still confused about the number of stratification groups being generated. For instance, in p.4 line 137 the authors inform about nine education level groups. Then in Table 2 (p.4) only three groups are reported (1-6, 7-12, 12<). Then in supplementary Tables 1, 2 and 3, and 4 and 5, 30 and 25 groups respectively, stratified by both age and education are reported. Nowhere in the manuscript those stratifications are appropriately justified statistically. Over-stratification may involve at least two problems. On the one hand, it may undermine the representativity of percentile scores in specific groups constituted by a small number of individuals (smaller than 50, see Mitrushina et al 2005). On the other hand, an artificially increase of number of subgroups that doesn’t exhibit real differences in performance will produce irrelevant norms for clinicians. Accordingly, the authors should justify and describe in more detail the criteria being used to divide subgroups withing each demographic variable (education and age) statistically. Reviewing preceding procedures and criteria being used in the literature (for Spanish speaking and non-Spanish speaking populations) may help to justify or improve their stratification in the most appropriate way.

In this regard, the description of MANOVA results in Table 3 seems insufficient. A more detailed description of these statistical analyses should be provided in the text specifying degrees of freedom, sphericity values and corrections… Moreover, Post-hoc analyses describing between-group differences between consecutive groups (age groups and education groups) should be described. The use of such a systematical stratification procedure will help generating a more precise and robust set of norms.

Minor points

Reflections about the construct validity of verbal fluency tasks as reported in P.2 (lines 49-63) should be better connected to the existing literature about it. Otherwise, some asserts may result speculative. Also, the sentence “However, the cognitive processes involved in the evaluation remain the same.” (p.2 line 50) is a matter of current debate given the plausible asymmetrical difficulty (and consequently attention, executive functions, etc) naming words with a given letter across languages with a different number of words starting with this letter. The authors should recognize in their introduction the fact that no Spanish versions of the phonological or semantic fluency tasks have been validated in this language jet.

P.4 line 134-135 May be the term “instruction” should be replaced by the term “education” as defined above

The specific instructions provided to subjects to perform the fluency test should be specified in the “Testing procedure” section. If instructions correspond to any already described procedure, please, include the reference.

The number of decimal points should be homogeneous along tables. Given the nature of the dependent variable (number of words) two decimals seems enough to describe mean and SD values. Review Tables 1, 2 and 3

The ordering of fluency variables within tables should be equal across them (for instance, see differences between Table 4 and 5).

Author Response

Dear Reviewer,

Thanks for your observations. To clarify the description of our sample, we include a specific paragraph about why we selected those group ranges (related to the educational profiles in Peru) and a description of the method used to score phonological and semantic fluency. We include a paragraph to complete some information (lines 65-71).

About percentiles, we include an appendix. Considering the number of groups by sex, age, and educational level, we initially just used descriptive statistics (supple tables).  Our first intention was to publish the percentiles separately, now are all included here in this manuscript. 

Sincerely

Reviewer 2 Report

Hello,

Congratulation, 

You wrote a good article include clearly information about your goals and methods. 

I have a suggestion for you, because you imply to Verbal Fluency in different languages, so please point to other researches and articles which have been done in some different culture and countries like Arabic and Persian .

Author Response

Dear Reviewer,

Thanks for your observations. We include a paragraph to complete information about your suggestion (lines 65-71).

Sincerely

Reviewer 3 Report

Dear Authors I read you work entitled "Verbal Fluency Test: Validation and Normative data on Peruvian Population" from pediatric to old age." and here I enclose my recommendations to you: 

1. In the Introduction section it will be good to make a reference paragraph for VFT in other languages and also clinical population. That will help the Introduction rational.

2. In the Methods section have a sub-chapter on the scoring system for VFT that was used, since there are different suggested scoring methods in the literature.

Thank you. 

Author Response

Dear Reviewer,

Thanks for your observations. To clarify the description of our sample, we include a specific paragraph about why we selected those group ranges (related to the educational profiles in Peru) and a description of the method used to score phonological and semantic fluency. We include a paragraph to complete some information about other verbal fluency and languages (lines 65-71).

Sincerely

Round 2

Reviewer 1 Report

I want to thank the authors for their responses to my questions that, in my opinion, have contributed to clarify their manuscript. However, there is still a key point that still needs to be solved as mentioned in my previous review.

I still have serious doubts about the necessity, utility, and representativity of all the groups being stratified to generate the present norms. The rationale applied to divide among different Education groups is now clearer. However, there is still a problem justifying the numerosity of age groups. As previously mentioned, over-stratification may involve at least two problems. On the one hand, it may undermine the representativity of percentile scores in specific groups constituted by a small number of individuals (smaller than 50, see Mitrushina et al 2005). On the other hand, an artificially increase of number of subgroups that doesn’t exhibit real differences in performance will produce irrelevant norms for clinicians. Accordingly, the authors should justify and describe in more detail the criteria being used to divide subgroups withing each demographic variable (education and age) statistically. Reviewing preceding procedures and criteria being used in the literature (for Spanish speaking and non-Spanish speaking populations) may help to justify or improve their stratification in the most appropriate way.

Post-hoc analyses describing between-group differences between consecutive groups (age groups and education groups) represent a possible systematical stratification procedure that will help generating a more precise and robust set of norms.

Author Response

Dear reviewer,

Thank you for your comments. We check our sample groups seeing small groups between the following age ranges:  26-30,31-40, 41-50, and 71 to 95, but many of them have more participants than fifty (50). Later, we analyze the groups by sex, age range, and education.

We compared them on different phonological and semantic tasks and determined that many of these groups were not significantly different. Through post-hoc analyses, we found that the early age range and older participants have different performances, but most adults do not show differences. 

Nonetheless, regarding verbal fluency performance, it is completely normal and expected these profiles in control participants. We still believe in avoiding uniting the groups, especially considering that the focus for clinicians is the possibility of comparing to specific age range, sex, and education of non-control participants (expected psychiatric patients, for example), by the same demographic characteristic with the control group.

Besides, we considered at the beginning not to publish the appendix with the percentiles, but now, we hope MDPI could agree to join the appendix to the publication, leaving clinicians the possibility to compare specific demographics of clinic patients with controls on the Peruvian population.